# Exploring how and why attributes of existing and emerging early cancer detection tests influence experiences and participation among individuals at risk of socioeconomic disadvantage: A qualitative interview study

Stefanie Bonfield *, Evangelos Katsampouris , Suzanne E. Scott, Stephen W. Duffy, Fiona M. Walter, Samantha L. Quaife

Centre for Cancer Screening Prevention and Early Diagnosis, Wolfson Institute of Population Health, Queen Mary University of London, London, United Kingdom

* s.bonfield@qmul.ac.uk

## Abstract

### Background

Emerging sample biomarker tests promise to improve early cancer detection, but also stand to influence socioeconomic inequalities in uptake of asymptomatic screening and symptomatic referrals.

### Objective

To explore how and why different attributes of early cancer detection tests influence experiences and participation across test modalities and contexts among individuals at risk of socioeconomic disadvantage.

### Method

Qualitative semi-structured interviews with 30 individuals (aged ≥50 years) at risk of socioeconomic disadvantage explored how and why different attributes of early cancer detection tests affected their experiences and participation across test modalities and contexts. Vignette test scenarios and a think-aloud protocol were used to facilitate interviewees' deliberation between a range of test attributes. Data were analysed using framework analysis.

### Results

Select attributes of early cancer detection tests prevented equitable access due to conflicts with lived experiences such as caring responsibilities, reduced mobility, concerns about being stigmatised, and not being physically or psychologically able to undergo procedures. Participants perceived more invasive forms of testing conducted

**Data availability statement:** The interview transcripts contain potentially identifiable and sensitive health information. Participants did not consent for their transcripts to be shared publicly and therefore remain confidential under the approval of Queen Mary Ethics of Research Committee. Anonymised verbatim quotes are presented in the manuscript and supplementary files to illustrate the findings. Requests for access to the data can be submitted to Queen Mary Ethics of Research Committee research-ethics@qmul.ac.uk (ref: QMERC22.318), for researchers who meet the criteria for access to confidential data.

**Funding:** This work was supported by Bart's Charity (G-001522; MGU0461). SES is supported by Barts Charity (G-001520; MRC&U0036). The funding body was not involved in the conduct of the study or preparation of the manuscript.

**Competing interests:** The authors have declared that no competing interests exist.

by a doctor in a hospital to indicate greater risk of a cancer diagnosis, and higher perceived risk of cancer was associated with preferences for attributes perceived to deliver an accurate or quick result. Participants varied in what they considered a 'cancer test' and some did not want to test due to the perceived or experienced burden of waiting for or receiving a cancer result.

## Conclusions

Emerging sample tests to support early cancer detection could address existing barriers to uptake by offering greater convenience. Inequitable uptake may persist if individuals do not perceive there to be a sufficient reason to test, are not confident to take part or doubt the ability of sample tests to accurately detect signs of cancer.

---

## Introduction

Over 40% of individuals with cancer in the United Kingdom (UK) are diagnosed at an advanced stage [1]. Early cancer detection offers more promising prevention and treatment options and greater chances of survival. However, approximately 29–38% of eligible individuals do not participate in asymptomatic cancer screening programmes [2], and over 5% of individuals do not attend urgent symptomatic referral appointments [3]. Individuals experiencing poorer socioeconomic conditions make up a greater proportion of these groups who do not attend [3,4], yet face greater incidence and lower survival rates across several types of cancer [5,6]. It is crucial to consider the ways that early cancer detection tests are offered and delivered to provide more equitable opportunities to take part.

The COM-B model and the integrated screening action model (I-SAM) postulate that individuals need the capability (such as the knowledge of how to test, self-efficacy, physical mobility, financial resources), opportunity (such as the convenience, access, social norms and support) and motivation (based on previous experiences, habits, perceived risk and beliefs about harms and benefits) to take part in cancer testing [7,8]. Evidence suggests that individuals experiencing socioeconomic deprivation face additional barriers to taking part, including negative perceptions of cancer testing such as the invasiveness of tests (perceived threat), lower confidence in their ability to take part (self-efficacy), and fatalistic beliefs about cancer and the potential for curative treatment (response efficacy) [9–11]. However, adapting attributes of existing cancer tests has improved participation or attitudes towards testing among those who previously chose not to take part [12–14].

Preference research can guide test developers, policy makers and healthcare professionals in delivering early detection tests in ways that align with public and patient needs and optimise uptake. In turn this can reduce the burden of late diagnosis on patients and healthcare systems. This is important given that several new test modalities to support early cancer detection are under development, including sample tests such as blood, breath and urine tests to detect cancer biomarkers of multiple different types of cancer [15]. These novel tests could be used to triage individuals invited to

undergo more invasive forms of testing in symptomatic and asymptomatic pathways [16]. It is important to evaluate the potential impact and promise of these cancer tests on inequalities in uptake within early cancer detection pathways before (as well as during and after) their implementation, as emphasised by the CanTest framework [17].

Existing public and patient studies have commonly explored preferences for attributes of a single type of cancer test or groups of tests delivered through a single cancer pathway such as cancer screening programmes [18,19]. This makes their results less applicable to emerging modalities which could be implemented at different stages and within more than one early cancer detection pathway. In addition to experiencing barriers to engaging with cancer tests, individuals experiencing socioeconomic deprivation also face additional challenges to taking part in health research [20]. Consequently, there have been calls to ensure that targeted efforts are made to include these groups in future research to narrow cancer disparities [21,22]. A holistic approach to examining preferences among this population for attributes that are relevant across different modalities and cancer pathways is needed to inform equitable implementation of new early detection tests. This qualitative interview study used a vignette task and think-aloud protocol to explore how and why attributes of existing and emerging early cancer detection tests influence testing experiences and participation among those at risk of socioeconomic disadvantage in England.

## Materials and methods

### Study design and setting

Qualitative semi-structured interviews were conducted and supplemented with written descriptions of cancer detection tests, a vignette task (where hypothetical test scenarios were presented) and an adapted think-aloud protocol (where participants were encouraged to deliberate between different test attributes while thinking out loud). Interview materials were developed in consultation with two public and patient representatives. The study was designed and conducted using a critical realist approach [23,24]. It was assumed that observable attributes of early cancer detection tests are perceived and experienced differently depending on an individual's subjective reality and lived experiences. Ethical approval was granted by Queen Mary Ethics of Research Committee (01/02/2023: QMERC22.318). The research was conducted in compliance with relevant guidelines and with respect to participants' privacy. All participants provided written informed consent except for one participant who preferred to provide audio-recorded verbal consent documented using an encrypted Dictaphone (approved by the ethics committee in advance). The methods and results are reported according to the consolidated criteria for reporting qualitative research (COREQ checklist), see S1 Table [25]. The study protocol is available in S2 File.

### Recruitment and sample

People living in London were recruited via community-based methods between 1st February 2023 and 1st February 2024. Community organisations (such as community centres, social housing associations, public health authorities, libraries, sports centres, food banks, large employers, and organisations supporting community outreach initiatives) were approached in areas associated with greater deprivation to assist with recruitment. Individuals were invited to take part in the study directly either by a researcher (SB) or community gatekeeper at community groups and events, or indirectly using flyers in community venues and online adverts posted in local WhatsApp (by community gatekeepers) and Facebook (by the researcher) groups.

Participants were sampled purposively. Individuals who expressed interest in taking part were asked to provide information to check that they were eligible before arranging an interview based on the following inclusion criteria: 1) Aged 50 years or older (an age group that accounts for 90% of new cancer cases [26], and is likely to have already been or will soon be invited for cancer screening); 2) Living in the top 30% most deprived postcodes nationally (measured by the Index of Multiple Deprivation (IMD) 2019 which ranks residential postcodes according to seven domains of deprivation [27]); 3) Met an individual-level measure of socioeconomic disadvantage (measured by low household income [28], social

housing tenure [29,30], or a current or most recent manual or routine occupation [31]); 4) No previous cancer diagnosis in the last five years; 5) Good understanding of written and spoken English. Information about participants' gender and ethnicity were monitored using a sampling matrix to inform the recruitment strategy and maximise the diversity of the sample.

The aim was to recruit approximately 30 individuals to capture a broad range of experiences and attitudes among the target population based on considerations of information power and previous qualitative studies exploring related topics with similar target populations [32–35].

**Interview materials**

A semi-structured topic guide was developed consisting of questions about individuals' attitudes towards specific attributes of cancer detection tests. These related both to tests they had previous experience making decisions about (i.e., had been invited to consider before or had taken part in the test before) and/or cancer tests that might be offered to them in the future. Questions were designed to be open-ended (e.g., 'What was your experience of being able to get to the test?') to allow individuals to reflect on how attributes influenced their decision to test or their experiences (e.g., preferences for a test location) and/or why they held those views (e.g., preferred a local location because they were less mobile). Interview questions and prompts were loosely based on the COM-B model of behaviour change [7], to understand how and why attributes influenced their capability, opportunity and motivation to take part in testing. The subsequent analysis interpreted interviewees' accounts to identify attributes that appeared to be instrumental in driving test decision-making and participation.

To help individuals consider which attributes of tests would be important, particularly those without previous experiences of being invited to take part in testing, two sets of interview materials were designed to introduce individuals to attributes of existing and emerging cancer detection tests. This included example test descriptions and a vignette task. Vignette tasks are a contemporary research tool that involve presenting participants with a hypothetical scenario to elicit individuals' opinions and preferences [36]. It was expected that an individual's first impressions and deliberations between test options would highlight which attributes were important for decision making and why, as well as which, if any, attributes were considered irrelevant or unimportant. For this reason, individuals were encouraged to read each set of materials and express their first impressions, thought processes and feelings out loud while engaging with each task, known as an adapted think-aloud protocol [37].

The first set of materials included descriptions of five different types of tests that may be used to find signs of cancer: sample tests (e.g., blood, breath, urine, faeces or saliva tests), physical and visual inspection tests, imaging tests (e.g., CT scans, x-rays, MRI scans, ultrasound), biopsy and endoscopy. These were adapted from the NHS website and described briefly what each type of test involves from a patient perspective [38–40].

The second set of materials included two vignette tasks where individuals were asked to place themselves in the position of a fictional individual at risk of cancer. In one vignette, an individual was described as experiencing fatigue and weight loss. In another, a different individual was asymptomatic but had been identified at risk of cancer based on their age and family history. Each vignette included four test options that varied by different attributes such as the location of the test, whether it would be comfortable and how long it would take to receive the results. Attributes were informed by a narrative review and stakeholder consultations with experts in early cancer detection testing including six healthcare professionals, seven academic experts, four test technology/pathway experts, three public health professionals, one policy expert and one cancer research funder. Vignette tasks are appropriate for facilitating the discussion of sensitive topics [36]. It was expected that the inclusion of vignette scenarios described in the third person would allow individuals to share their preferences for attributes of tests without feeling obliged to share personal experiences or reasons for taking part in health testing if they did not wish to.

The interview topic guide including interview materials and the think-aloud protocol were used flexibly depending on participants' experiences with tests and preferred topics of discussion and are available in S3 File.

## Public and patient involvement

Two public and patient involvement representatives with previous experience of taking part in cancer testing were recruited through existing contacts to inform the design and conduct of the study. They reviewed the topic guide and interview materials for acceptability, task difficulty, comprehension and timing and suggested methods for recruiting participants. They were not involved in the analysis or interpretation of the findings.

## Data collection

Interviews took place between 10th March 2023 and 8th February 2024 and were conducted in person in private rooms in community settings close to where participants lived. The interviewer (SB) was a female behavioural science PhD student with previous experience conducting qualitative interviews. No one else was present. Participants were previously unknown to the research team. All participants were told that the aim of the research was to improve how tests are offered. The researcher made it clear at the beginning of each interview that they did not work for the National Health Service or have any clinical training and could not provide medical advice. Each interview was audio recorded using an encrypted Dictaphone.

After reading the participant information sheet and providing verbal or written informed consent, participants were asked to fill in a short, optional survey about their socio-demographic information. Participants were given a choice of whether to fill this in themselves or for the researcher to ask them the questions and record their responses. The audio-recording was then switched on and participants were asked questions about their previous experiences of having or being invited to have tests for cancer and which attributes of the tests they would find off-putting if they were invited to have a test in future.

Participants were asked to read the descriptions of different types of early cancer detection tests and to think aloud while considering if anything about each test could be potentially off-putting from a patient perspective. Then participants were asked to engage with the vignette task and to think-aloud while arranging the tests in order of preference that they thought the individual in the vignette would choose. The interviewer (SB) offered to read written materials aloud to participants if they preferred. When shown examples of different types of cancer tests, participants were informed that tests were not always used to look for cancer and may be used in many different medical contexts. During the vignette tasks, participants were told that they could look at one test option at a time if they preferred and did not have to examine all test options if they did not wish to. Lastly, participants were asked about any other aspects of testing that they felt would be important if they were offered a cancer test in the future.

After each interview, the interviewer (SB) noted down initial impressions and field notes of key factors influencing each participant's experiences and engagement with tests. The final order of the vignette test options was recorded for each participant. Participants were given a debrief sheet including a reminder of the aims of the research and contact details of support services in case they were negatively affected by the sensitive nature of the discussions or desired medical advice. All participants were provided with a £25 Love2Shop voucher to thank them for their contribution.

## Data analysis

The data were analysed in NVivo (version 12) using framework analysis and guided by seven recommended stages: Transcription, Familiarisation, Coding, Developing a working analytical framework, Applying the analytical framework, Charting the data into a framework matrix, Interpreting the data [41,42]. This method does not dictate whether data is analysed inductively or deductively and allows this to be informed by the research topic [42]. This was considered appropriate for systematically identifying similarities and differences in attitudes towards attributes of tests before interpreting thematic associations [41–43]. Other analytical approaches which are predominantly informed by the data (e.g., Grounded theory) were not selected because the researchers were interested in exploring attitudes towards pre-defined attributes of cancer

detection tests. The focus of the analysis was to explore which attributes contributed to negative experiences or attitudes towards taking part in testing and why. It was expected that the influence of attributes on experiences and participation would both be relevant to future engagement with tests. The COM-B model was used as a lens to explore the influence of attributes across experiences and participation. Analysis of responses to the third person vignette tasks was focused on the reasons individuals cited for preferred attributes based on their own beliefs *(e.g., 'I think most people would prefer')* or personal circumstances *(e.g., 'If it was me..').* The analysis also considered the asymptomatic and symptomatic context of the vignette being discussed.

Interview recordings were transcribed verbatim and personally identifiable data omitted by a professional transcription company. Three members of the research team (SB, EK, SLQ) familiarised themselves with and independently coded 10% of the data inductively before meeting to discuss initial impressions and a potential coding framework. SB then applied the analytical framework to the remaining transcripts, meeting regularly with the rest of the research team to discuss the suitability of the framework and make appropriate revisions.

The research team held substantial knowledge and previous research experience of qualitative methods, behavioural science and health psychology, cancer screening and cancer diagnosis in primary care. FMW is also a qualified General Practitioner (GP). SB took notes during meetings with the research team to discuss insights and reflections and kept a reflexive diary throughout data collection and analysis. The reflexive diary was revisited several times during analysis to inform iterations of data organisation and interpretation. In addition to the COM-B model [7], the I-SAM [8] was used to arrange the data into sub-themes describing factors related to attributes of early cancer detection tests that influenced individuals' capability, opportunity and motivation to take part. The I-SAM was considered useful for informing how sub-themes should be arranged and labelled given its outline of factors influencing screening behaviour specifically, including but not limited to the attributes of tests. Participants who consented were provided with a summary of the findings.

## Results

### Sample characteristics

Approximately 200 individuals were invited to take part in the study directly by the researcher (SB) in community centres. The number of individuals invited to take part using indirect recruitment methods such as through flyers, social media adverts and invitation by community members is unknown. Of the 166 individuals who expressed interest in taking part and answered the eligibility questions, thirty-nine (23.5%) were eligible based on the inclusion criteria. Nine did not take part; three were uncontactable and six decided not to take part due to being too busy (n = 3), changing their mind (n = 2) or not wishing to consent (n = 1).

Interviews were conducted with 30 individuals (20 females) living across 11 London boroughs. The interviews lasted a mean duration of 35 minutes (SD = 5.7, range: 23–45). Participants were aged from 52 to 86, with a mean of 63.8 years (SD = 8.9). The majority (n = 22) had previous experience of having cancer tests and a few (n = 4) had received a previous cancer diagnosis over 5 years ago. Over half were from White ethnic backgrounds (n = 18), and some belonged to Asian (n = 2), Black (n = 4) and Mixed (n = 5) ethnic groups. Sample characteristics are shown in Table 1.

### Thematic framework

The thematic framework, shown in Fig 1, was developed by using the COM-B model to explore how and why attributes of early cancer detection tests appeared to influence participants' experiences and participation [7]. The first theme describes how and why test attributes were interpreted to prevent equitable access to early cancer detection tests by lowering individual's capability and opportunity to take part. The second theme presents an interpretation of how and why test attributes influenced participants' motivation to take part in testing. Sub-themes are supplemented with numbered illustrative quotes (e.g., Quote 1 is denoted by Q1). Further quotes are available in S4 Table. Case studies are presented

**Table 1. Sample characteristics.**

|  | N (%) |
|---|---|
| **Gender** |  |
| Female | 20 (66.7%) |
| Male | 9 (30.0%) |
| Prefer not to say | 1 (3.3%) |
| **Age** |  |
| 50–59 | 12 (40.0%) |
| 60–69 | 11 (36.7%) |
| 70–79 | 6 (20.0%) |
| 80–89 | 1 (3.3%) |
| Mean in years (range) | 63.8 (52, 86) |
| **Ethnicity** |  |
| Asian | 2 (6.7%) |
| Black | 4 (13.3%) |
| Mixed | 5 (16.7%) |
| White | 18 (60.0%) |
| Prefer not to say | 1 (3.3%) |
| **IMD 2019 decile** |  |
| 1 – most deprived | 1 (3.3%) |
| 2 | 24 (80.0%) |
| 3 | 5 (16.7%) |
| **Socioeconomic indicators** |  |
| Low household income | 24 (80.0%) |
| Social housing tenure | 21 (70.0%) |
| Routine or manual occupation | 10 (33.3%) |
| **Employment status** |  |
| Employed | 5 (16.7%) |
| Unemployed | 14 (46.7%) |
| Retired | 11 (36.7%) |
| **Education** |  |
| Finished school < 16 | 7 (23.3%) |
| GCSE or equivalent | 6 (20.0%) |
| A'level or equivalent | 6 (20.0%) |
| Degree | 11 (36.7%) |
| **Previous cancer diagnosis** |  |
| No | 26 (86.7%) |
| Yes | 4 (13.3%) |
| **Health literacy** *(support needed reading medical information)* |  |
| Never | 19 (63.3%) |
| Rarely | 7 (23.3%) |
| Sometimes | 4 (13.3%) |
| **Self-reported health status** |  |
| Poor | 2 (6.7%) |
| Fair | 11 (36.7%) |
| Good | 10 (33.3%) |
| Very Good | 6 (20.0%) |

*(Continued)*

**Table 1.**  (Continued)

| | N (%) |
|---|---|
| Excellent | 1 (3.3%) |
| **Previous cancer test experience** | |
| No | 8 (26.7%) |
| Yes | 22 (73.3%) |

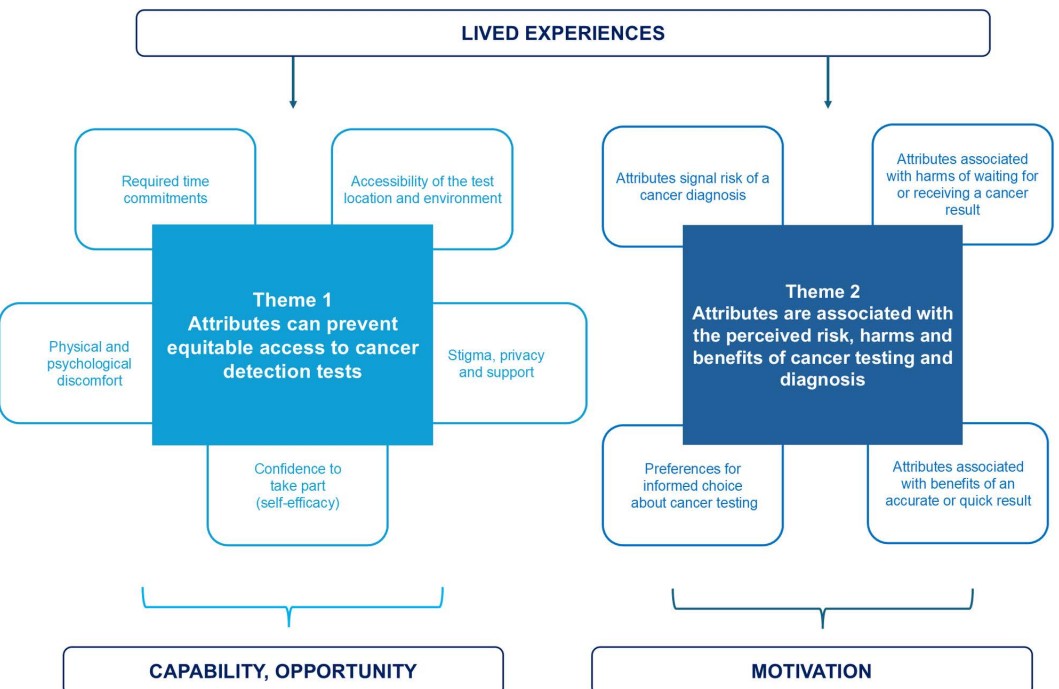

**Fig 1.  Thematic framework.** Themes are shown in boxes with surrounding sub-themes. Lived experience is illustrated as a cross-cutting theme.

to illustrate the cross-cutting theme of lived experiences and their importance when considering the influence of test attributes on participation.

### Theme 1: Attributes can prevent equitable access to cancer detection tests

Individuals discussed several attributes of early cancer detection tests that influence their capability and opportunity to take part based on their individual circumstances. This included the required time commitments, accessibility of the test location and environment, stigma, privacy and support, confidence to take part and physical and psychological discomfort.

**Required time commitments.**  For some, the time required to take part in testing was inconvenient due to competing commitments such as work, caring responsibilities and childcare that they could not afford to miss (Q1). Participants voiced concerns about the duration beyond the test procedure itself, including time spent getting to the location and in the waiting room. Participants associated hospital locations with requiring more time to attend than mobile units and GP surgeries.

Q1: *"I was looking after my mum who was bedridden, also my little boy's special needs.... I can't really leave him very much on his own because he is autistic… I've got no family over here …. and I can't go just pay a babysitter £20, it's just not possible on part time wages and a single parent with the cost of living"* Female, age 50–59, Mixed

**Accessibility of the test location and environment.** Many said they would prefer a 'local' test and a particular venue due to the location in relation to where they live, often citing GP surgeries or mobile units. Participants discussed the importance of needing accessible transport options such as nearby car parking spaces or bus stops and accessible venues, particularly for those who are less mobile with one individual highlighting that stairs on mobile units are 'not easy to get up and down'. Test locations that were not easy to access were cited by a few participants as the reason for not attending screening appointments (Q2).

Participants also discussed the importance of the test environment. While one participant described the hospital environment as 'cold', another described mobile units as 'uncomfortable' and 'cramped'. One participant also disliked being exposed to 'uncontrolled loud noise' in waiting rooms, stating that it could reduce their likelihood of taking part.

Q2: *"I didn't, haven't been --Which is naughty, but I don't know, it seemed a lot more effort to get there than it used to"* Female, age 60–69, White

**Stigma, privacy and support.** Many expressed concerns that mobile units parked in a supermarket made testing more visible to others (Q3). Some were worried about the lack of privacy in the waiting room or test room and liked the privacy afforded by self-sample tests conducted at home.

Many participants described the importance of receiving information, support and reassurance from healthcare professionals and family members when engaging with cancer testing. Concerns about being stigmatised by healthcare professionals were commonly discussed for tests that involved undressing such as cervical screening, mammograms and colonoscopies (Q4). A few preferred a healthcare professional of the same sex to carry out tests to minimise embarrassment. Some described previous cancer testing as a dehumanising experience and recounted feeling like 'a piece of meat'. Participants explained that this feeling was exacerbated by a lack of compassion from the person conducting the test and called for more empathy and communication.

Q3: *"People's lining up and you might see someone that you don't want them seeing you or thinking, oh, he's gone for a cancer scare or something."* Female, age 70–79, White

Q4: *"Some people are very negative on, when they are examining you. And it's sort of like, you know, we can't all walk in here a size 10."* Female, age 60–69, White

**Confidence to take part (self-efficacy).** Many participants had positive views of tests that were considered 'standard' and 'normal' or 'to be expected'. Participants preferred attributes of tests that were 'familiar' such as their local GP surgery or knowing the individual healthcare professional doing the test. Many said that familiarity with testing due to previous experiences made subsequent testing easier.

Many participants raised concerns about having the confidence to complete self-sample testing correctly, particularly when testing for cancer, and the worry about whether they had 'done it right'. Participants held positive attitudes towards self-sample tests that were 'easy to do'. Although, some highlighted that individuals who are less mobile may need the support from a relative (Q5).

Those who described the challenges of engaging with tests valued the chance to ask questions and a choice of how test procedures were carried out to accommodate personal circumstances.

Q5: *"Especially if you're not mobile enough to get out of your house, how are you going to do your testing at home safely and accurately?"* Male, 50–59, White

**Physical and psychological discomfort.** Many described the anticipated or experienced pain and invasiveness of testing as off-putting as well as the potential for side effects and complications. Participants explained that getting older

and physical changes increase the pain and discomfort they experienced during cervical and breast cancer testing. Some said that they could 'not tolerate' it anymore so had stopped attending, particularly if they knew friends who had also decided to stop (Q6). Some participants discussed phobias such as needle phobia and claustrophobia which contributed to negative experiences of tests, particularly for MRI scans (Q7). A few said they would only take part in more invasive tests in future if they were offered sedation.

Q6: *"I don't want to go back, because my friend also said that it's painful and then she doesn't want to go any more."* Female, age 50–59, Asian

Q7: *"You don't know what that person has been through in the past … then if you're going to get stuck in the tunnel for the MRI scan, it's giving me flashbacks of being confined and scared."* Female, age 50–59, Mixed

**Theme 2: Attributes are associated with the perceived risk, harms and benefits of cancer testing and diagnosis**

Individuals reflected on attributes in relation to their motivation to take part. Some explained that motivation was influenced by the perceived risk of a cancer diagnosis, which in some cases was based on the reason they were offered a test or the type of test involved. Some did not want to take part in testing due to the harms associated with attributes of testing such as the burden of waiting for or receiving a cancer result. Others were inclined to test when cancer risk was high and prioritised attributes that were associated with greater accuracy and/or shorter time to receiving a diagnosis, even when these were unfavourable in other ways. Some wanted clearer information to make an informed decision.

**Attributes associated with perceived risk of a cancer diagnosis.** Participants associated attributes of tests with the risk of a cancer diagnosis. Some described sample tests as 'general tests' in contrast to biopsy, endoscopy and imaging tests, as well as tests conducted in a hospital or by a doctor, which were perceived as 'more serious' and as indications that 'there's something wrong with you' (Q8). Some described low perceived risk of having cancer as a reason for not taking part such as when they were not experiencing symptoms or if they had recently undergone symptomatic investigations where they received a clear result (Q9).

Q8*: "The fact that the test is done by a doctor rather than say a nurse or a nurse practitioner suggests it's a bit more serious."* Male, 50–59, White

Q9: *"Bowel cancer test kit came in the post. I didn't ask for it, I don't know why it came. … my half-sister's had bowel cancer ….. just because she's had it doesn't mean I'm going to have it… I just didn't feel I wanted to engage with that…. I think I more or less threw it away."* Age 60–69

**Attributes associated with harms of waiting for or receiving a cancer result.** Anxiety and fear about being diagnosed with cancer were commonly reported in the lead up to testing, during the test and when waiting for the result; most participants preferred to receive the results quickly. Some described the anxiety caused by being called for follow-up tests because 'your head goes to cancer straight away'. Participants described experiences or expectations of the emotional or financial consequences of receiving a cancer diagnosis and the burden that it might inflict on friends and family, particularly if they had caring responsibilities (Q10).

Q10: *"I wouldn't want to know if I had got cancer, because then I'd have to tell my family, and I don't know whether I could do that. Not right now while my mum's still alive, because she's got her own health issues and stuff …. she'd just get too upset."* Female, age 50–59, White

**Attributes associated with benefits of an accurate or quick cancer result.** Many participants were motivated to test if there was a higher risk of cancer being found. In this situation, many preferred attributes that were associated

with perceived accuracy. Some participants were concerned about the accuracy of self-sample tests and a few were concerned about the impact of false positive results such as needing a further unnecessary test 'for nothing'. For many, the hospital was perceived as the best place to get tested because it is a 'medical environment' and is where the 'cancer specialists' and 'proper tests' are (Q11). These attributes were associated with being 'further along the line' and an accurate diagnosis was preferred to facilitate early detection and treatment. A few believed it was worth waiting longer for an appointment or for test results, or tolerating a painful test if it meant they got a 'proper result' (Q12). Some individuals preferred attributes that they perceived would shorten the time to receiving a diagnosis and starting treatment such as tests that could be conducted or deliver results more quickly.

Q11: *"especially like a, something like cancer, you just, I do take it straight to the professionals."* Male, age 50–59, Black

Q12: *"If you feel uncomfortable when the test is being done, if it's, if we're talking about cancer, then I guess you put up with it, don't you?"* Female, age 60–69, White

**Preferences for informed choice about cancer testing.** Participants varied in what they discussed when asked about cancer testing. When asked if they had ever been invited for and/or had a test for cancer, some reflected on previous experiences of symptomatic investigations for cancer or attending cancer screening programmes. Others asked 'does screening count?' or initially reported that they had not experienced cancer testing before, but subsequently discussed previous experiences of cancer screening or being referred for investigations. Some described experiencing frustration in the past when they were not informed about 'what they're looking for' and wanted it to be made explicit if cancer concern was a reason for being offered a test. Participants wanted to make an informed decision about testing by being given clear information about the about what a test would involve, and the harms and benefits (Q13).

Q13: *"I'd really like to know all the steps involved in terms of what the test is, how long it will take, when I'd get the results and all that information …. is my particular demographic more susceptible to a particular type of cancer and that's why I'm being offered the test? Is it a mass screening?"* Female, age 70–79, Mixed

### Lived experiences as a cross-cutting theme

It was common for participants to describe links between their lived experiences and the impact of multiple test attributes on their experiences and engagement with cancer tests over time. Two case studies are presented in Table 2 to illustrate the relationship between lived experiences and the role of test attributes in preventing equitable access (Theme 1) and influencing the perceived risk, harms and benefits of taking part (Theme 2).

## Discussion

This qualitative study with individuals at risk of socioeconomic disadvantage indicated that certain attributes of early cancer detection tests can prevent equitable access due to conflicts with lived experiences. Emerging tests to support early cancer detection such as blood, breath and urine sample tests could improve individuals' capability and opportunity to take part by requiring fewer time commitments (e.g., due to less travel required to locations, shorter procedures and potentially more varied appointment times), being available in more accessible locations and environments (e.g., community based settings), offering greater familiarity and privacy (e.g., not requiring individuals to remove clothing, although community settings may cause concerns about being seen by others), and imposing fewer physical and psychological demands (e.g., compared to physical and psychological discomfort associated with imaging, endoscopy and biopsy tests). This may be particularly welcomed by individuals who have had negative previous experiences with more invasive tests

**Table 2. Lived experiences as a cross-cutting theme.**

| Lived experience | Sub-theme and illustrative quotes |
|---|---|
| **Case study 1** | |
| Previous experiences of sexual abuse, a fear of needles and claustrophobia meant that invasive testing is a psychologically traumatic experience. The participant valued the test being located somewhere nearby and the staff being emotionally supportive and accommodating. | <u>Physical and psychological discomfort</u><br>"I'm also one of these people who's suffered a lot of rape and sexual abuse … so I have problems touching my body … let alone to have somebody else grabbing hold of it to such a degree."<br>"It's helpful to have machines which are away from them a bit, because too enclosed, somebody like me panic and it won't get done." |
| | <u>Accessibility of the test location and environment</u><br>"I don't see any reason why we can't get a bus to drop us right off at the hospital …… because when emotions is flying you need a safer way home." |
| | <u>Stigma, privacy and support</u><br>"I felt like I was handled like some piece of meat of some form, and even though I could see that, well, she didn't appear horrible, the nurse that carried it out, but this grabbing of the breasts and shoving it and wanting to flatten and putting it underneath this thing, and it hurt." |
| **Case study 2** | |
| Receiving an incorrect terminal cancer diagnosis had a negative financial impact. This caused anxiety when subsequently called for an abnormality found in a screening programme which was exacerbated by a lack of compassion from the staff conducting the test. The participant did not think current symptoms were likely to mean there was something wrong due to previous inaccuracies. They were also reluctant to impose the burden of another cancer diagnosis on their husband. | <u>Stigma, privacy and support</u><br>[mammogram experience] "It's like, "yeah, there's an abnormality on it" … I'm shaking like a leaf. There was no compassion, there was no, "sit down, how do you feel about it?" I know they're very busy but it was, "you're a lump of meat." |
| | <u>Attributes associated with harms of waiting for or receiving a cancer result</u><br>"They say this is it, you've got three years, I've been in that room, that's my future….. I will say I was reckless afterwards. I did empty my bank account, … it was like, what I've been saving for my old age. Now I'm having to make up for it... I blew my money."<br>"I won't say it's a lump, I've got a sore bit… Not going to complain.. to enquire about things because you think, "if they do find something, it's probably nothing but am I going to worry myself into it?"<br>"If he ever had to be with me for another one of those diagnoses, someone saying, "oh, she's got cancer and you're going to lose her" again, I wouldn't want him to go through that." |

conducted in the traditional hospital setting. It is likely that these findings extend to those who face similar barriers to participation regardless of socioeconomic disadvantage, underlining the value of achieving equitable access to improve uptake across the population.

The findings are corroborated by previous research exploring barriers to engagement with cancer detection testing including cancer stigma [44], transport issues [11], and caring responsibilities [11]. The results also support previous studies which have reported positive attitudes towards less invasive sample tests for estimating cancer risk and for multi-cancer early detection alongside concerns about test accuracy [45,46]. In our study, individuals were asked to consider and compare attributes related to various types of cancer detection tests. When comparing different testing contexts, the convenience of taking part was not always a priority when individuals considered the potential risk of a cancer diagnosis, which some inferred from attributes of the test. For example, more invasive forms of testing conducted by a doctor in a hospital setting were associated with a greater risk of there being something wrong.

In support of theoretical models of fear responses and protection motivation theory [47,48], when perceived risk or salience of a cancer diagnosis was high, participants evaluated the effectiveness of testing in dealing with the threat. Some were reluctant to do a self-sample test for cancer; they were concerned about doing the test correctly

(self-efficacy) and a few doubted the accuracy and diagnostic capability of sample tests (response-efficacy), preferring a test conducted by a doctor in a hospital to ensure an accurate diagnosis. This suggests that the increased accessibility of sample tests for cancer detection may not lead to greater motivation to take part if individuals doubt the value of a test such as its importance or accuracy. Alternatively, tests may be taken up but induce greater concerns about false negatives and frustration about not being eligible for referral. It is not clear from this study whether these perceptions are more prominent among those at risk of socioeconomic disadvantage or also persist across the population.

The findings raise an important question about how emerging early detection tests for single and multiple types of cancer would be introduced to and perceived by eligible individuals. Participants seemed to vary in what they considered to be a test for cancer. This ambiguity is intuitive given that many tests used in early cancer detection pathways are used to detect physiological changes that are risk factors for several health conditions and diseases besides cancer. Additionally, previous research has found that GPs did not always make the reason for referral to a cancer pathway explicit to symptomatic patients [49]. However, the association of sample tests with cancer may become more explicit in future if innovations lead to improvements in test accuracy and detection capability. This suggests that during the implementation of novel, less invasive tests, it will be important to communicate the aims of testing, the accuracy of the results and implications for further testing and diagnosis to support informed decision making. Further research will be needed to understand the perceived risk of a cancer diagnosis associated with sample tests used to rule out single and multiple types of cancer in specific contexts such as risk stratification and triage, and the most effective methods of communicating cancer risk information.

### Strengths and limitations

This study focused on the perspectives of individuals at risk of socioeconomic disadvantage; the majority had low household income (80%), lived in social housing (70%) and were living in the top 20% most deprived postcodes nationally (83%), aligning with NHS England's CORE20 approach to narrowing health inequalities [50]. However, there are some limitations. Firstly, all participants were aged 50 or over and living in urban settings. Preferences among younger individuals and those living in rural settings may differ, as has been found in previous research [51,52]. Individuals with lower educational background, those in employment and individuals from Asian backgrounds were also relatively under-represented in the sample, despite efforts to approach community organisations with links to these groups.

Secondly, participants were asked to arrange hypothetical test options in order of preference from the perspective of a fictional individual. In practice, an individual would be offered a single test and could decide not to take part. Consequently, individuals may have been reluctant to disclose negative attitudes towards cancer detection testing. It is also possible that stated preferences for specific attributes were influenced by other attributes that appeared within the hypothetical test descriptions. Importantly, there is limited evidence for the validity of behavioural intentions reported in vignette tasks in relation to actual behaviour [53]. Further work is needed to understand which attributes relevant to emerging sample test modalities would influence real world participation.

### Conclusions

Attributes of early cancer detection tests present numerous barriers to taking part for individuals at risk of socioeconomic disadvantage, who report inequitable experiences and uptake due to varied lived experiences and personal circumstances including caring responsibilities, reduced physical capability, psychological trauma and concerns about being stigmatised. While emerging early cancer detection tests may offer more accessible opportunities to take part, inequitable uptake may persist if individuals do not perceive there to be a sufficient reason to test, are not confident to take part or doubt the accuracy of sample tests and their ability to facilitate diagnosis and treatment.

## Supporting information

**S1 Table. Study methods and results reporting according to COnsolidated criteria for REporting Qualitative research (COREQ) checklist.**
(DOCX)

**S2 File. Protocol.**
(DOCX)

**S3 File. Interview materials.**
(DOCX)

**S4 Table. Illustrative quotes.**
(DOCX)

## Acknowledgments

The authors would like to thank Dr Georgia Black for her advice and expertise during the design of the interview materials and Margaret Ogden and David Holden (public and patient representatives) for providing valuable feedback for refining the interview topic guide and materials. They would also like to thank the staff from local community centres who offered their help in recruiting individuals to take part in the study and provided private spaces for the interviews to be held in.

## Author contributions

**Conceptualization:** Stefanie Bonfield, Stephen W. Duffy, Fiona M. Walter.

**Data curation:** Stefanie Bonfield.

**Formal analysis:** Stefanie Bonfield, Evangelos Katsampouris.

**Funding acquisition:** Samantha L. Quaife.

**Investigation:** Stefanie Bonfield.

**Methodology:** Stefanie Bonfield, Evangelos Katsampouris, Stephen W. Duffy, Fiona M. Walter, Samantha L. Quaife.

**Project administration:** Stefanie Bonfield.

**Software:** Stefanie Bonfield, Samantha L. Quaife.

**Supervision:** Evangelos Katsampouris, Suzanne E. Scott, Stephen W. Duffy, Fiona M. Walter, Samantha L. Quaife.

**Validation:** Stefanie Bonfield, Evangelos Katsampouris, Suzanne E. Scott, Stephen W. Duffy, Fiona M. Walter, Samantha L. Quaife.

**Visualization:** Stefanie Bonfield, Evangelos Katsampouris, Suzanne E. Scott, Stephen W. Duffy, Fiona M. Walter, Samantha L. Quaife.

**Writing – original draft:** Stefanie Bonfield.

**Writing – review & editing:** Evangelos Katsampouris, Suzanne E. Scott, Stephen W. Duffy, Fiona M. Walter, Samantha L. Quaife.

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
