## [Decision Letter · Decision Letter 0]

PONE-D-25-05025Exploring how and why attributes of existing and emerging early cancer detection tests influence experiences and participation among individuals at risk of socioeconomic disadvantage: a qualitative interview studyPLOS ONE

Dear Dr. Bonfield,

Thank you for submitting your manuscript to PLOS ONE. After careful consideration, we feel that it has merit but does not fully meet PLOS ONE’s publication criteria as it currently stands. Therefore, we invite you to submit a revised version of the manuscript that addresses the points raised during the review process.

We look forward to receiving your revised manuscript.

Kind regards,

Veincent Christian Pepito

Academic Editor

PLOS ONE

Journal Requirements:

Additional Editor Comments:

Dear authors, thank you very much for a well-written manuscript. However, the reviewers have identified some issues which I hope you will address.

Reviewers' comments:

Reviewer's Responses to Questions

**Comments to the Author**

1. Is the manuscript technically sound, and do the data support the conclusions?

Reviewer #1: Yes

Reviewer #2: Yes

2. Has the statistical analysis been performed appropriately and rigorously? 

Reviewer #1: N/A

Reviewer #2: N/A

3. Have the authors made all data underlying the findings in their manuscript fully available?

Reviewer #1: No

Reviewer #2: Yes

4. Is the manuscript presented in an intelligible fashion and written in standard English?

Reviewer #1: Yes

Reviewer #2: Yes

5. Review Comments to the Author

Reviewer #1: Dear Authors,

The investigation of users’ preferences regarding cancer detection tests among people at risk of socioeconomic disadvantage can contribute to overcoming barriers to access those tests. Please see my comments below.

When you respond to the comments, please provide your answers in a table with two columns, one with comments and the other with your answers. In the answer column, for each comment separately, include the revised text, in addition to your answer.

“Over 40% of individuals in the United Kingdom (UK) are diagnosed with cancer at an advance stage.(1)” Do you mean that 40% of the population is sick or that 40% of the diagnosed people are already in an advanced stage? Please revise the sentence to improve clarity.

Please correct the citation style. Insert the citation in the appropriate place: with the associated text, not after it (before “.” or “,”, not after them).

Please explain why it is important to know people’s preferences. It is important to motivate the usefulness of your study.

Who cares about people’s preferences? From a supply-demand perspective, when the supply (tests) is much lower compared to the demand (need for tests), the supply side doesn’t care what the consumers (patients) want. If there are 100 pairs of shoes on the market and 1 million barefoot people needing shoes, who cares what color they like or even what size?

Also, mention who should consider these preferences (scientists who create the tests, professionals who use it on patients, etc.).

Why did you focus only on people at risk of socioeconomic disadvantage? Do they represent the majority of sick people? What about the rest? Are they unimportant?

Can the results be extrapolated to people who are not at risk? Why/ Why not?

Please explain why the vignette task and think-aloud protocol are suitable for this study.

Please mention what data analysis methods you used and motivate your choice.

“This qualitative study aimed to use a vignette task and think…” The interview is not mentioned. Please add it. Also, it should be written that the study “used” instead of “aimed to use”.

“Qualitative semi-structured interviews supplemented with written descriptions of cancer detection tests, vignettes of tests scenarios and an adapted think-aloud protocol as participants were encouraged to deliberate between different test attributes” The predicate is missing. Please revise.

“People living in London were recruited via community-based methods between 01/02/2023 – 08/02/2024.” It is not clear what the month and the day are. Please use words for the months.

Please briefly explain what a vignettes task and a think-aloud protocol are.

Please explain what framework analysis is and why this was considered suitable and preferred to other methods for qualitative data.

“Participants were asked to read the descriptions of different types of early cancer detection tests and to think aloud while considering if anything about each test could be potentially off-putting from a patient perspective. Then participants were asked to engage with the vignette task and to think aloud while arranging the tests in order of preference that they thought the individual in the vignette would choose…. ” In addition to this description, please provide the interview script, the list with all questions used in the interview. Include the main questions, helping questions, etc. When presenting the questions, indicate the specific objective to which they respond. For example, it is mentioned that the study objective is to find out “how and why attributes of …. tests influence testing experiences and participation”. Show what questions focused on “how”, “why”, “experiences”, and “participation”.

This information supports the replication of the study.

“FMW is also a qualified GP.” What does GP mean?

The text in Fig is very blurred. Please improve it.

“Individuals discussed several attributes of early cancer detection tests that influence their capability and opportunity to take part based on their individual circumstances…..(the rest of the text)”. Vignettes are about imaginary people, Alex and Sam. The manuscript text presents the interview results as they were about the interview participants. This creates confusion. Please provide the necessary information to remove the confusion. How can you know they referred to themselves when they answered the questions about Alex and Sam and not to some imaginary people? What did you do to ensure that?

What do Q1, Q2, .. mean? Q from quote, or is this a code for each person? Please explain.

“Although, some highlighted that individuals who are less mobile may need the support from a relative (5).” Does “(5)” refer to the quote mentioned below this sentence? If yes, please write Q5.

The study objective is to find out “how and why attributes of …. tests influence testing experiences and participation”. Please present the results in a way that responds to the how and why, and highlights the experiences, and participation.

Reviewer #2: Thank you for the opportunity to review this manuscript, which is timely regarding the likely future direction of cancer early detection, and in considering it from the perspective of those (inverse care law) least served by current cancer screening/early detection (or with the most to potentially benefit).

It was great to see the involvement of public/patient members in the research material development - were they involved in any of the analysis or interpretation (I think from the coding description that they were not?). Was any other health literacy lens used for the materials given the target audience?

Thank you for the COREQ this was also useful.

I wasn't familiar with Think-Aloud Protocols - it would be useful if the description and a little more detail on this was provided at first use of the term.

It wasn't clear to me why participants needed a debrief sheet (line 164) - this is usually used in studies which involve deception?

It wasn't clear to me why framework analyses was used over other types of qual analyses (line 168) - another sentence or two justification would be helpful to the reader.

Appropriate limitations were outlined including the nature of the sample being relatively highly educated, urban, and white/english speaking.

I have two broad questions to the authors to consider whether any further clarification/detail in the manuscript may be useful:

The I-SAM (which includes COM-B) and CanTest are about informing/designing equitable screening/integrated programmes - from the protocol it appears the research wasn't originally designed using either framework - were these selected as part of the presentation of results? COM-B (the third part of I-SAM) appears to be the main aspect used in the results for this paper. Thinking about inequities, SES inequities as the focus of this work, the I-SAM proposes ways that theory and research can improve screening access in particular via patient and environmental factors - the background and the findings/discussion indicate a strong attention to the participant/patient factors but less on the opportunity side (particularly the environmental aspect/framing - opportunities to design programmes/approaches using these findings eg improving systems health literacy/information accessibility, how tests are offered and who by, ways to improve trustworthiness etc.

The second question was about including both screening (asymptomatic) and early detection (symptomatic) vignettes - do you think there was any confusion about this for the participants, and whether the symptomatic vignette impacted the perspectives/preferences more? Were the responses analyses separately for the different vignettes?

It was interesting that this study was about a broad range of test types across a broad set of potential screening/early detection settings - however one key note in the discussion (line 413) was that further research will be needed to understand perceived risk associated with tests in specific contexts. Given the potential nature of MCEDs/panels used for risk stratification/eligibility/screening/early detection do the authors think that studies in context will be possible?

6. PLOS authors have the option to publish the peer review history of their article (what does this mean? ). If published, this will include your full peer review and any attached files.

**Do you want your identity to be public for this peer review?** For information about this choice, including consent withdrawal, please see our Privacy Policy .

Reviewer #1: No

Reviewer #2: No

---

## [Author Response · Author response to Decision Letter 1]

15 May 2025

Dear Veincent Christian Pepito,

We are very grateful to you and the reviewers for taking the time to review the paper and provide meaningful feedback. We have addressed the journal requirements including updating the file labels, manuscript format and data availability statement. We have also changed the reference numbered 45 (a report published by Cancer Research UK) to the more recently published and associated journal article. The reference numbered 48 has been updated to an alternative publication as the previous article had been redacted.

We have now addressed each comment from the reviewers below and believe these revisions have significantly strengthened the clarity, replicability and impact of the study. We hope that you will now consider this acceptable for publication in PLOS ONE.

Yours sincerely,

Stefanie Bonfield, MSc

PhD student, Centre for Cancer Screening, Prevention, and Early Diagnosis, Wolfson Institute of Population Health, Queen Mary University of London

Dr Samantha Quaife, PhD CPsychol

Reader in Behavioural Science, Centre for Cancer Screening, Prevention, and Early Diagnosis, Wolfson Institute of Population Health, Queen Mary University of London

No. Comment Response

Reviewer#1

1. Dear Authors, The investigation of users’ preferences regarding cancer detection tests among people at risk of socioeconomic disadvantage can contribute to overcoming barriers to access those tests. Please see my comments below.

When you respond to the comments, please provide your answers in a table with two columns, one with comments and the other with your answers. In the answer column, for each comment separately, include the revised text, in addition to your answer.

R1: Thank you very much for your helpful comments, please see your responses laid out below as you request.

2. “Over 40% of individuals in the United Kingdom (UK) are diagnosed with cancer at an advance stage.(1)” Do you mean that 40% of the population is sick or that 40% of the diagnosed people are already in an advanced stage? Please revise the sentence to improve clarity.

R2: Thank you, this has been revised for clarity:

Over 40% of individuals with cancer in the United Kingdom (UK) are diagnosed at an advanced stage.(1)

3. Please correct the citation style. Insert the citation in the appropriate place: with the associated text, not after it (before “.” or “,”, not after them).

R3: This has now been amended for all in-text citations.

4. Please explain why it is important to know people’s preferences. It is important to motivate the usefulness of your study.

Who cares about people’s preferences? From a supply-demand perspective, when the supply (tests) is much lower compared to the demand (need for tests), the supply side doesn’t care what the consumers (patients) want. If there are 100 pairs of shoes on the market and 1 million barefoot people needing shoes, who cares what color they like or even what size?

Also, mention who should consider these preferences (scientists who create the tests, professionals who use it on patients, etc.).

R4: Thank you, the benefit of preference research for those who receive tests as well as the healthcare systems involved in their delivery has been outlined in the introduction as follows:

Preference research can guide test developers, policy makers and healthcare professionals in delivering early detection tests in ways that align with public and patient needs and optimise uptake. In turn this can reduce the burden of late diagnosis on patients and healthcare systems. This is important given that several new test modalities to support early cancer detection are under development …

5. Why did you focus only on people at risk of socioeconomic disadvantage? Do they represent the majority of sick people? What about the rest? Are they unimportant?

R5: Thank you, the increased cancer risk and mortality facing these groups has been outlined in the first paragraph:

Individuals experiencing poorer socioeconomic conditions make up a greater proportion of these groups who do not attend(3, 4), yet face greater incidence and lower survival rates across several types of cancer (5, 6).

The reason for focusing on this group based on their under-representation in previous research is now reiterated in the final paragraph of the introduction:

In addition to experiencing barriers to engaging with cancer tests, individuals experiencing socioeconomic deprivation also face additional challenges to taking part in health research.(20) Consequently, there have been calls to ensure that targeted efforts are made to include these groups in future research to narrow cancer disparities.(21, 22)

6. Can the results be extrapolated to people who are not at risk? Why/ Why not?

R6: Thank you, we have now acknowledged the validity of extending the findings beyond the population in the discussion section for the first theme:

It is likely that these findings extend to those who face similar barriers to participation regardless of socioeconomic disadvantage, underlining the value of achieving equitable access to improve uptake across the population.

And the second theme:

It is not clear from this study whether these perceptions are more prominent among those at risk of socioeconomic disadvantage or also persist across the population.

7. Please explain why the vignette task and think-aloud protocol are suitable for this study.

R7: Thank you, the reasons for including the vignette task and think-aloud protocol have now been outlined in the ‘Interview Materials’ section:

To help individuals consider which attributes of tests would be important, particularly those without previous experiences of being invited to take part in testing, two sets of interview materials were designed to introduce individuals to attributes of existing and emerging cancer detection tests. This included example test descriptions and a vignette task. Vignette tasks are a contemporary research tool that involve presenting participants with a hypothetical scenario to elicit individuals’ opinions and preferences.(36). It was expected that an individual’s first impressions and deliberations between test options would highlight which attributes were important for decision making and why, as well as which, if any, attributes were considered irrelevant or unimportant. For this reason, individuals were encouraged to read each set of materials and express their first impressions, thought processes and feelings out loud while engaging with each task, known as an adapted think-aloud protocol(36).

8. Please mention what data analysis methods you used and motivate your choice.

R8: Thank you, this has been expanded in the Data analysis section as follows:

The data were analysed in NVivo (version 12) using framework analysis and guided by seven recommended stages: Transcription, Familiarisation, Coding, Developing a working analytical framework, Applying the analytical framework, Charting the data into a framework matrix, Interpreting the data(41, 42). This method does not dictate whether data is analysed inductively or deductively and allows this to be informed by the research topic.(41) This was considered appropriate for systematically identifying similarities and differences in attitudes towards attributes of tests before interpreting thematic associations.(40-42) Other analytical approaches which are predominantly informed by the data (e.g. Grounded theory,) were not selected because the researchers were interested in exploring attitudes towards pre-defined attributes of cancer detection tests.

9. “This qualitative study aimed to use a vignette task and think…” The interview is not mentioned. Please add it. Also, it should be written that the study “used” instead of “aimed to use”.

R9: Thank you for highlighting this, the sentence now reads:

‘This qualitative interview study used a vignette task and think-aloud protocol to explore…’

10. “Qualitative semi-structured interviews supplemented with written descriptions of cancer detection tests, vignettes of tests scenarios and an adapted think-aloud protocol as participants were encouraged to deliberate between different test attributes” The predicate is missing. Please revise.

R10: Thank you, this now reads as follows:

‘Qualitative semi-structured interviews were conducted and supplemented with written descriptions of cancer detection tests, a vignette task (where hypothetical tests scenarios were presented) and an adapted think-aloud protocol (where participants were encouraged to deliberate between different test attributes while thinking out loud).’

11. “People living in London were recruited via community-based methods between 01/02/2023 – 08/02/2024.” It is not clear what the month and the day are. Please use words for the months.

R11: This has been amended:

‘People living in London were recruited via community-based methods between 1st February 2023 and 1st February 2024.’

12. Please briefly explain what a vignettes task and a think-aloud protocol are.

R12: The ‘Study design and setting’ section now reads as:

‘Qualitative semi-structured interviews were conducted and supplemented with written descriptions of cancer detection tests, a vignette task (where hypothetical tests scenarios were presented) and an adapted think-aloud protocol (where participants were encouraged to deliberate between different test attributes while thinking out loud).’

This is further explained in the ‘Interview Materials’ section, see comment #7.

13. Please explain what framework analysis is and why this was considered suitable and preferred to other methods for qualitative data.

R13: Thank you, this has been expanded in the Data analysis section as follows:

The data were analysed in NVivo (version 12) using framework analysis and guided by seven recommended stages: Transcription, Familiarisation, Coding, Developing a working analytical framework, Applying the analytical framework, Charting the data into a framework matrix, Interpreting the data(41, 42). This method does not dictate whether data is analysed inductively or deductively and allows this to be informed by the research topic.(41) This was considered appropriate for systematically identifying similarities and differences in attitudes towards attributes of tests before interpreting thematic associations.(40-42) Other analytical approaches which are predominantly informed by the data (e.g. Grounded theory,) were not selected because the researchers were interested in exploring attitudes towards pre-defined attributes of cancer detection tests.

14. “Participants were asked to read the descriptions of different types of early cancer detection tests and to think aloud while considering if anything about each test could be potentially off-putting from a patient perspective. Then participants were asked to engage with the vignette task and to think aloud while arranging the tests in order of preference that they thought the individual in the vignette would choose…. ” In addition to this description, please provide the interview script, the list with all questions used in the interview. Include the main questions, helping questions, etc. When presenting the questions, indicate the specific objective to which they respond. For example, it is mentioned that the study objective is to find out “how and why attributes of …. tests influence testing experiences and participation”. Show what questions focused on “how”, “why”, “experiences”, and “participation”.

This information supports the replication of the study.

R14: Thank you, the topic guide, including questions and prompts, has now been included in the supplementary file alongside the interview materials and instructions for the think-aloud protocol. We hope this improves the replicability of the study.

Questions were designed to be open-ended to allow individuals to reflect on the influence of attributes on their decisions to take part or their experiences of taking part, both of which were expected to be relevant to how and why attributes influence participation. This is now described in the interview materials section using examples to illustrate these objectives:

A semi-structured topic guide was developed consisting of questions about individuals’ attitudes towards specific attributes of cancer detection tests. These related both to tests they had previous experience making decisions about (i.e., had been invited to consider before or had taken part in the test before) and/or cancer tests that might be offered to them in the future. Questions were designed to be open-ended (e.g. ‘What was your experience of being able to get to the test?’) to allow individuals to reflect on how attributes influenced their decision to test or their experiences (e.g. preferences for a test location) and/or why they held those views (e.g. preferred a local location because they were less mobile). Interview questions and prompts were loosely based on the COM-B model of behaviour change(7), to understand how and why attributes influenced their capability, opportunity and motivations to take part in testing. The subsequent analysis interpreted interviewees’ accounts to identify attributes that appeared to be instrumental in driving test decision-making and participation.

15. “FMW is also a qualified GP.” What does GP mean?

R15: Thank you this has been clarified as a General Practitioner (GP).

16. The text in Fig is very blurred. Please improve it.

R16: Thank you, this has been addressed and the revised figure passes checks in PLOS’s NAAS tool and the PACE digital diagnostic tool. We believe the image may remain blurred in the PDF builder of the submission form, but is not when the image is downloaded.

17. “Individuals discussed several attributes of early cancer detection tests that influence their capability and opportunity to take part based on their individual circumstances…..(the rest of the text)”. Vignettes are about imaginary people, Alex and Sam. The manuscript text presents the interview results as they were about the interview participants. This creates confusion. Please provide the necessary information to remove the confusion. How can you know they referred to themselves when they answered the questions about Alex and Sam and not to some imaginary people? What did you do to ensure that?

R17: Thank you for raising this. We have now provided further information about the justification for including a third person vignette task in the ‘Interview materials’ section:

Vignette tasks are appropriate for facilitating the discussion of sensitive topics.(36) It was expected that the inclusion of vignette scenarios described in the third person would allow individuals to share their preferences for attributes of tests without feeling obliged to share personal experiences or reasons for taking part in health testing if they did not wish to.

We have also described the steps taken to ensure the results were reflective of participants’ personal preferences when discussing the vignette tasks in the ‘Data analysis’ section:

Analysis of responses to the third person vignette tasks was focused on the reasons individuals cited for preferred attributes based on their own beliefs (e.g. ‘I think most people would prefer’) or personal circumstances (e.g. ‘If it was me ..’).

18. What do Q1, Q2, .. mean? Q from quote, or is this a code for each person? Please explain.

R18: This has now been clarified in parentheses:

Sub-themes are supplemented with numbered illustrative quotes (e.g. Quote 1 is denoted by Q1).

19. “Although, some highlighted that individuals who are less mobile may need the support from a relative (5).” Does “(5)” refer to the quote mentioned below this sentence? If yes, please write Q5.

R19: Thank you for highlighting this error, this has now been amended to (Q5).

20. The study objective is to find out “how and why attributes of …. tests influence testing experiences and participation”. Please present the results in a way that responds to the how and why, and highlights the experiences, and participation.

R20: Thank you for raising this ambiguity. The reasons for analysin

---

## [Decision Letter · Decision Letter 1]

Exploring how and why attributes of existing and emerging early cancer detection tests influence experiences and participation among individuals at risk of socioeconomic disadvantage: a qualitative interview study

PONE-D-25-05025R1

Dear Dr. Bonfield,

We’re pleased to inform you that your manuscript has been judged scientifically suitable for publication and will be formally accepted for publication once it meets all outstanding technical requirements.

Kind regards,

Veincent Christian Pepito

Academic Editor

PLOS ONE

Additional Editor Comments (optional):

Reviewers' comments:

Reviewer's Responses to Questions

**Comments to the Author**

1. If the authors have adequately addressed your comments raised in a previous round of review and you feel that this manuscript is now acceptable for publication, you may indicate that here to bypass the “Comments to the Author” section, enter your conflict of interest statement in the “Confidential to Editor” section, and submit your "Accept" recommendation.

Reviewer #1: All comments have been addressed

Reviewer #2: All comments have been addressed

2. Is the manuscript technically sound, and do the data support the conclusions?

Reviewer #1: Yes

Reviewer #2: (No Response)

3. Has the statistical analysis been performed appropriately and rigorously? 

Reviewer #1: N/A

Reviewer #2: (No Response)

4. Have the authors made all data underlying the findings in their manuscript fully available?

Reviewer #1: No

Reviewer #2: (No Response)

5. Is the manuscript presented in an intelligible fashion and written in standard English?

Reviewer #1: Yes

Reviewer #2: (No Response)

6. Review Comments to the Author

Reviewer #1: Dear Authors,

You responded to my comments. I do not have new ones.

Good luck with your research!

Kind regards,

The Reviewer

Reviewer #2: (No Response)

7. PLOS authors have the option to publish the peer review history of their article (what does this mean? ). If published, this will include your full peer review and any attached files.

**Do you want your identity to be public for this peer review?** For information about this choice, including consent withdrawal, please see our Privacy Policy .

Reviewer #1: No

Reviewer #2: No

---

## [Editor Report · Acceptance letter]

PONE-D-25-05025R1

PLOS ONE

Dear Dr. Bonfield,

I'm pleased to inform you that your manuscript has been deemed suitable for publication in PLOS ONE. Congratulations! Your manuscript is now being handed over to our production team.

Kind regards,

on behalf of

Mr Veincent Christian Pepito

%CORR_ED_EDITOR_ROLE%

PLOS ONE